# CI-VAE: a Generative Deep Learning Model for Class-Specific Data Interpolation

## ABSTRACT

We propose a new variant of Variational Autoencoder (VAE), Class-Informed VAE (CI-VAE), that enables interpolation between arbitrary pairs of observations of the same class. CI-VAE combines the general VAE architecture with a linear discriminator layer on the latent space to enforce the construction of a latent space where observations from different classes are linearly separable. This allows for robust latent-space linear traversal and data generation between two arbitrary observations of the same class, which has potential applications in science and engineering. One specific application is to enhance understanding of the biological processes involving the development of diseases or cancer from healthy cells. We demonstrate the effectiveness of CI-VAE on the MNIST dataset of handwritten digits, showing that it significantly improves class-specific linear traversal and data augmentation compared to VAE while maintaining comparable reconstruction error. We also apply CI-VAE to a study of colon cancer single-cell genomics data, showing that interpolation between normal cells and tumor cells using CI-VAE may enhance our understanding of the mechanism of cancer development.

## 1 INTRODUCTION

Variational Autoencoders (VAEs) are unsupervised probabilistic neural network models Kingma & Welling (2013); Dilokthanakul et al. (2016); Hsu et al. (2017) that have gained popularity in a variety of applications, including computer vision Hsieh et al. (2018); Vahdat & Kautz (2020); Tabacof et al. (2016); Huang et al. (2018), natural language processing Wu et al. (2019); Bahuleyan et al. (2017); Semeniuta et al. (2017), genomics, and precision medicine Grønbech et al. (2020); Minoura et al. (2021). These models use an encoder to infer a probabilistic latent representation of data in a lower-dimensional space. While VAEs were originally designed for unsupervised learning, various variants have been developed for supervised and semi-supervised learning as well Connor et al. (2021); Joy et al. (2020); Mi et al. (2021); Kameoka et al. (2019); Kingma et al. (2014); Gómez-Bombarelli et al. (2018); Ehsan Abbasnejad et al. (2017); Xu et al. (2017); Wu et al. (2019); Sohn et al. (2015); Higgins et al. (2016); Zhao et al. (2019). VAEs are particularly useful for synthetic data generation and dimensionality reduction.

Suppose we intend to understand how two observations in high-dimensional space transform from one to another. We can use linear traversal in the latent space of a VAE to generate a trajectory and observe the transformation. This approach has many potential applications in engineering and science. One example is studying the transformation of healthy neural cells to cells with Parkinson's disease to uncover the disease mechanism Hook et al. (2018); Blauwendraat et al. (2020). In such a study, it is important to ensure that the latent space is linearly separable among different classes of cells (cell-types), so that linear traversal does not overlap between different classes.

To address this issue, we propose Class-Informed Variational AutoEncoders (CI-VAEs), a novel model architecture that includes an additional linear discriminator applied to the latent space. The linear discriminator is trained simultaneously with the rest of the model, and the classification loss is included in the overall cost function. Using the MNIST handwritten digit dataset LeCun et al. (1998), we demonstrate how CI-VAEs can perform class-specific linear traversal and class-specific synthetic data generation more robustly than VAEs. As a further example, we apply CI-VAEs to Colon Cancer single-cell RNAseq data Qi et al. (2022) to better understand the development of cancer cells from normal colon cells (see appendix).

## 2 AUTOENCODERS

Autoencoders are unsupervised neural network models that consist of an encoder and a decoder. The encoder maps data $x$ into a lower-dimensional latent space $z$, and the decoder reconstructs the data in its original form $\hat{x}$ (see Figure 1a). Autoencoders have a range of applications, including dimensionality reduction, data compression, and data denoising Simonyan & Zisserman (2014); Kramer (1991).

In autoencoders, the total cost function is the reconstruction error, which is the distance between the original input data $x$ and the reconstructed data $\hat{x}$. This is expressed as:

$$J_{AE} = \mathcal{L}(x, \hat{x}) \tag{1}$$

In contrast, variational autoencoders (VAEs) map the input data $x$ to a posterior distribution $p(z|x)$. Specifically, assuming a Gaussian distribution for $p(z|x)$, VAEs deterministically map the input data $x$ to the mean $\mu_{z|x}$ and standard deviation $\Sigma_{z|x}$ to construct the posterior distribution $p(z|x)$. A latent variable $z$ is then constructed by sampling from the posterior distribution. Finally, the decoder deterministically maps the latent variable $z$ back to the reconstructed input data $x$ An & Cho (2015); Kingma & Welling (2019).

The total cost function for VAEs consists of the reconstruction error and a regularization term that enforces the latent space distribution $p(z|x)$ to be close to the prior $p(z) = N(\vec{0}, \vec{I})$. This promotes orthogonality/independence of the $z$ axis and regularizes the network weights.

VAEs offer the advantages of projecting data into a lower-dimensional space, approximating the posterior distribution $p(z|x)$, and generating new synthetic data from the inferred latent space.

$$J_{VAE} = \mathcal{L}(x, \hat{x}) + \alpha \times KL[N(\mu_{z|x}, \sigma_{z|x}), N(\vec{0}, \vec{I})] \tag{2}$$

### 2.1 CLASS-INFORMED VARIATIONAL AUTOENCODERS (CI-VAE)

The proposed CI-VAE model is a supervised model based on the VAE architecture with additional components in both the architecture and the total cost function (see Figure 1c). To construct CI-VAE, we implemented the standard VAE architecture with the addition of a linear discriminator, which is a single-layer neural network that maps the latent variable $z$ to a set of probabilities for predicting its class $y$. The linear discriminator is trained with the rest of the network, and its trainable weights $W_{h \times d}$ are updated during training, where $h$ is the latent space dimension and $d$ is the number of unique classes $y$ in the data $x$. The classification loss (cross-entropy loss) of the linear discriminator is also included in the total loss of the VAE. This ensures that observations from different classes in the latent space $z$ are linearly separable. The linear discriminator has two simultaneous impacts: 1) during backpropagation, it becomes a better classifier, but its performance plateaus due to its linearity constraint, and 2) the gradient signals are sent to the entire model to infer a latent space that is linearly separable based on classes. The entire cost function of CI-VAE is therefore described as:

$$
\begin{aligned}
J_{CI-VAE} = \mathcal{L}(x, \hat{x}) + \alpha \times KL[N(\mu_{z|x}, \sigma_{z|x}), N(0,1)] + \\
\beta \times J_{LD}(y, \hat{y})
\end{aligned}
\tag{3}
$$

where $J_{LD}$ is the cross-entropy loss between the true class $y$ and the linear class $\hat{y}$ predicted by the linear discriminator for any observation $x$.

### 2.2 CLASS-SPECIFIC LINEAR TRAVERSAL AND CLASS-SPECIFIC DATA AUGMENTATION

Figure 2 shows a schematic representation of the latent space generated by CI-VAE. As illustrated in this figure, there is a higher probability of data around the origin $\vec{0}$. This is imposed by the KL-divergence loss between the posterior $p(z|x)$ and the prior $p(z) = N(\vec{0}, \vec{I})$. Additionally, observations from different classes are linearly separable due to the linear discriminator network in the CI-VAE model. Linear separability is desirable when generating within-class data through linear traversal in the latent space, because it theoretically guarantees that any within-class linear traversal

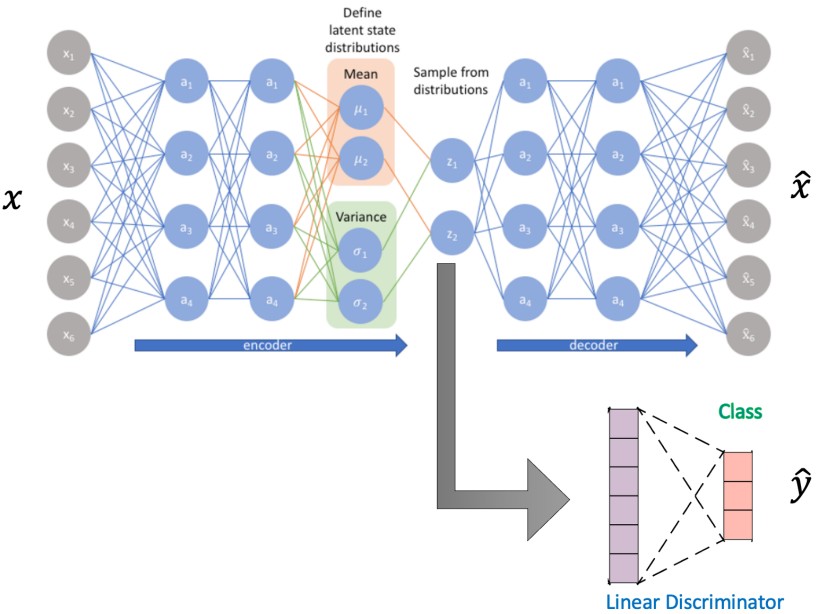

Figure 1: General architecture of Class-Informed Variational Autoencoders (CI-VAEs). The linear discriminator is a single linear layer neural network followed by a softmax function to predict class probabilities of observations on the latent space.

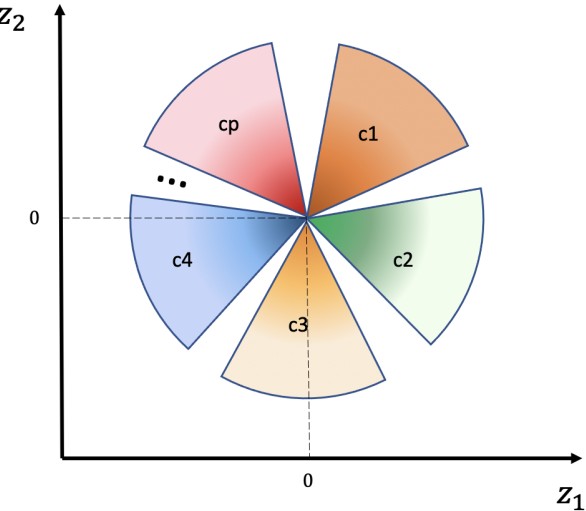

Figure 2: Schematic 2D representation of the latent space $z$ after trained on CI-VAE. Latent variables are normally distributed around 0 with the standard deviation of $I$ enforced by prior $p(z) = N(0, I)$. In addition, enforced by the linear discriminator in CI-VAE, classes are linearly separated.

will not interfere with the space of other classes. While there is often class overlap in VAEs, the geometrical property imposed on the latent space in CI-VAE minimizes the chance of interference during within-class data generation through linear traversal.

Linear traversal can be used for data augmentation in VAEs. CI-VAEs allow for class-specific linear traversal, which enables class-specific data augmentation or, in other words, the generation of synthetic labeled data. To do this, we repeatedly select pairs of observations on the latent space from the desired class and perform linear traversal while generating new samples from the same class.

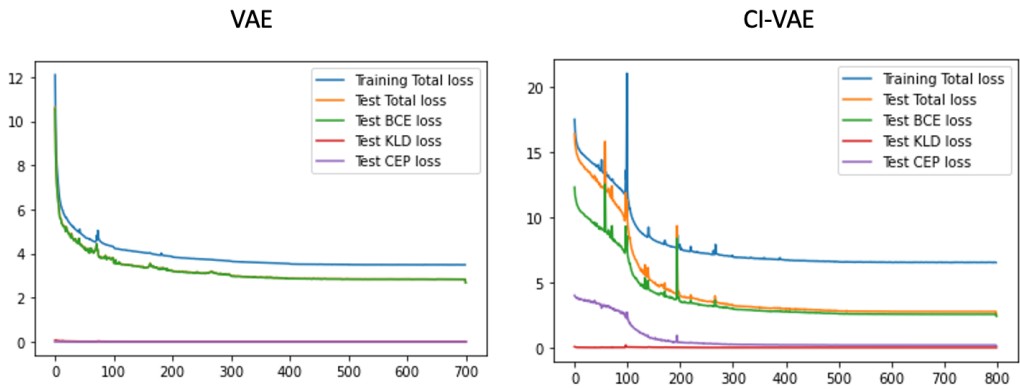

Figure 3: Average loss components v.s. epoch number for **Left**: VAE, and **Right**: CI-VAE

Table 1: Performance comparison between VAE and CI-VAE for reconstruction mean absolute error (MAE) and linear separability of classes on latent space $z$ measured by the performance of a linear logistic regression.

|  | VAE | CI-VAE |
|---|---|---|
| Reconstruction Error (MAE) | 2.82 | **2.56** |
| Linear Separability(%) | 90.81 | **99.87** |

## 3 EXPERIMENTS

We evaluated our proposed CI-VAE model on the MNIST handwritten digits dataset LeCun et al. (1998). In addition to training a CI-VAE model, we trained a standard VAE model as a benchmark. To ensure a fair comparison, we used the same model architecture (except for the linear discriminator in CI-VAE), the same hyperparameters, and the same training, validation, and test data.

The MNIST dataset consists of 80,000 images of handwritten digits, each with $28{\times}28{\times}1$ pixels, and the digits range from 0 to 9. We used the digits as the true class associated with each image, but it is possible to use any other arbitrary class in CI-VAE. Data processing included normalizing pixel values, flattening the features into a $28{\times}28{=}526$-dimensional space, and randomly splitting the data into 80% for the training set, 10% for the validation set, and the remaining 10% for the test set.

### 3.1 MODEL ARCHITECTURE

The encoder in the CI-VAE model consists of seven fully connected layers, each followed by batch normalization, a ReLU non-linear activation, and dropout. These layers map the input data to the mean and standard deviation vectors of the latent variable $z$ with 20 dimensions. A latent vector $z$ is then extracted by randomly sampling from a Gaussian distribution described by the mean and standard deviation vectors. The linear discriminator proposed in CI-VAE consists of a single fully connected neural network that takes the $z$ vector as input and predicts class probabilities $\hat{y}$ as output. The decoder also consists of seven fully connected layers with the same layout as the encoder, which deterministically maps from $z$ to the reconstructed input data $\hat{x}$. The weights in the model are trained for over 2000 epochs with a batch size of 512 images until convergence is reached (see Figure 3).

### 3.2 RESULTS

After training the CI-VAE and VAE models, we compared the following quantities for each model: 1) reconstruction quality, 2) linear separability of classes of observations in the latent space, and 3) quality of class-specific synthetic data generation through linear traversal.

We evaluated the reconstruction quality of the CI-VAE and VAE models using the mean absolute error (MAE) between the input data $x$ and the reconstructed data $\hat{x}$. The results, shown in Table 1,

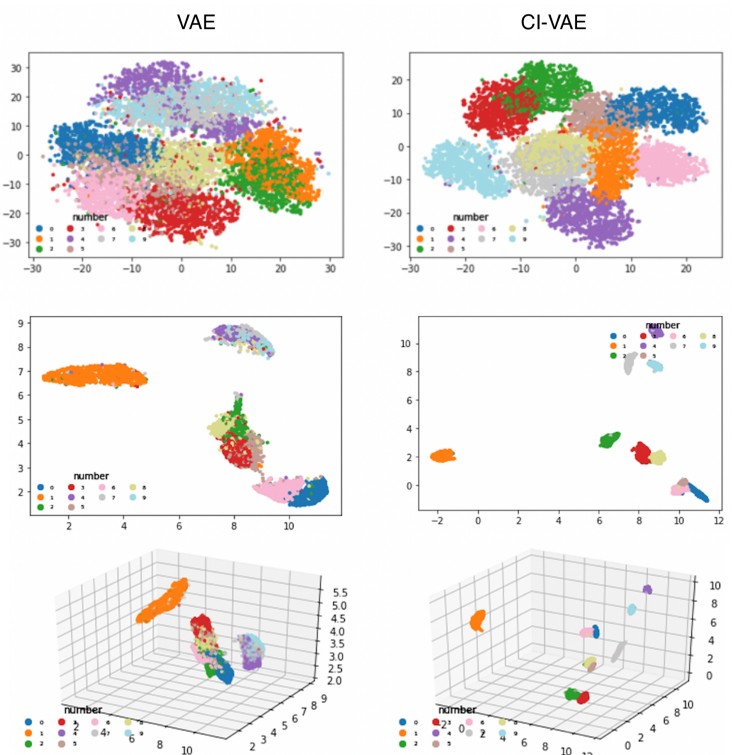

Figure 4: Representation of the latent space on test data for: **Top** : 2D TSNE plot, **Center** : 2D UMAP plot, **Bottom** : 3D UMAP plot for both VAE model on the Left and CI-VAE model on the Right.

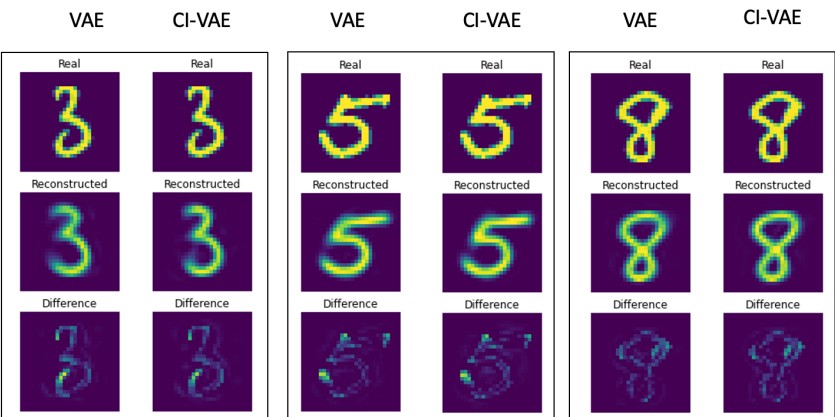

Figure 5: Top: a real observation, Middle: corresponding reconstructed observation, Bottom: error profile. Generated by VAE model (left) and CI-VAE model (right).

indicate that CI-VAE (with MAE = 2.56) outperformed VAE (with MAE = 2.82). To visualize the reconstruction performance of both models, we provided examples of real digit images, reconstructed images, and the difference images (defined as the simple subtraction of the real image from the reconstructed one) in figure 5. The difference images for both CI-VAE and VAE models are similar in appearance. We also assessed the linear separability of the classes in the latent space of each model. To do this, we trained a separate linear logistic regression classifier on the latent space representation of the test data, using it to predict the respective classes (digits in this case study) for both CI-VAE and VAE. The linear logistic regression represents a set of linear hyperplanes that best discriminate between classes in the latent space. The results, shown in Table 1, reveal that CI-VAE achieved a much higher classification accuracy (99.87%) compared to VAE (90.81 %), indicating that CI-VAE has greatly separated the classes in the latent space compared to VAE (by nearly 10% margin).

The enforcement of linear separation of classes in the latent space in CI-VAE improved the performance of within-class data generation through linear traversal. In figure 6, we demonstrated three examples of generating within-class data through latent space linear traversal in both CI-VAE and VAE models for digits 3, 5, and 8. As shown in the figure, VAE generated figures resembling different digits from the intended ones in each traversal. For instance, VAE generated an image of digit 8 during traversal within two digits 3 and generated images of digit 6 while traversing within two digits 5 and two digits 8. However, CI-VAE consistently generated samples of digits that were fully consistent with the intended digit in all experiments. This is because the classes in the latent space are linearly separated in CI-VAE, while they overlap significantly in VAE. While we only showed a few examples due to limited space, we can draw many more similar examples where CI-VAE robustly generates within-class data, but VAE fails to do so.

Within-class linear traversal can be used as a building block for within-class data augmentation. We generated synthetic data belonging to class $C$ by randomly selecting two points $a \in C$ and $b \in C$ on the latent space and generating new within-class synthetic data through linear traversal. Iterative random selection of pairs of points, linear traversal, and generating synthetic data allows for the generation of an unlimited number of within-class synthetic data. In situations where labeled data is limited, this method can be used in the CI-VAE model to generate a large amount of new labeled synthetic data. In figure 7, we used latent space linear traversal within digit 3 to generate many synthetic samples for digit 3. The data points in this figure are presented using a 2D TSNE plot of the original coordinates of the test data.

## 4 DISCUSSION

We developed CI-VAE to improve upon VAE by allowing for class separability in the latent space, which is the lower-dimensional representation of the data. We believed that using a linear discriminator to enforce the linear separability of the classes would address the issue of class overlap and allow for "class-specific" linear traversal and data interpolation.

In figure 1, we intentionally used a discriminator with no non-linearity to obtain a linearly separable latent space. One might suggest using a non-linear discriminator instead, but we chose a linear discriminator for several reasons: i) linear discriminators in the latent space are equivalent to non-linear high-dimensional hyperplanes in the original data space $x$; ii) when the linear discriminator is saturated and unable to linearly separate the data in the latent space, gradient backpropagation can send signals to the rest of the network to create a more linearly separable latent space while maintaining high reconstruction quality; iii) linear separability of classes allows for within-class linear traversal and thus to generate high-quality class-specific data generation.

By visualizing the latent spaces inferred by VAE and CI-VAE in figure 4, we can see that CI-VAE clearly separates the classes while VAE exhibits significant overlap among classes. It's worth noting that CI-VAE's ability to separate classes would be even more pronounced for more nuanced class definitions, such as assigning class 1 to female hand-written digits and class 2 to all other digits. CI-VAE can be applied to any class definition.

By adding a new term to the cost function of the model from the linear discriminator, as shown in equation 3, the reconstruction quality of the model may be affected. As shown in table 1, CI-VAE resulted in a higher reconstruction quality compared to VAE. This may be due to the inclusion of class information, which could have improved the overall performance of the model. However, it

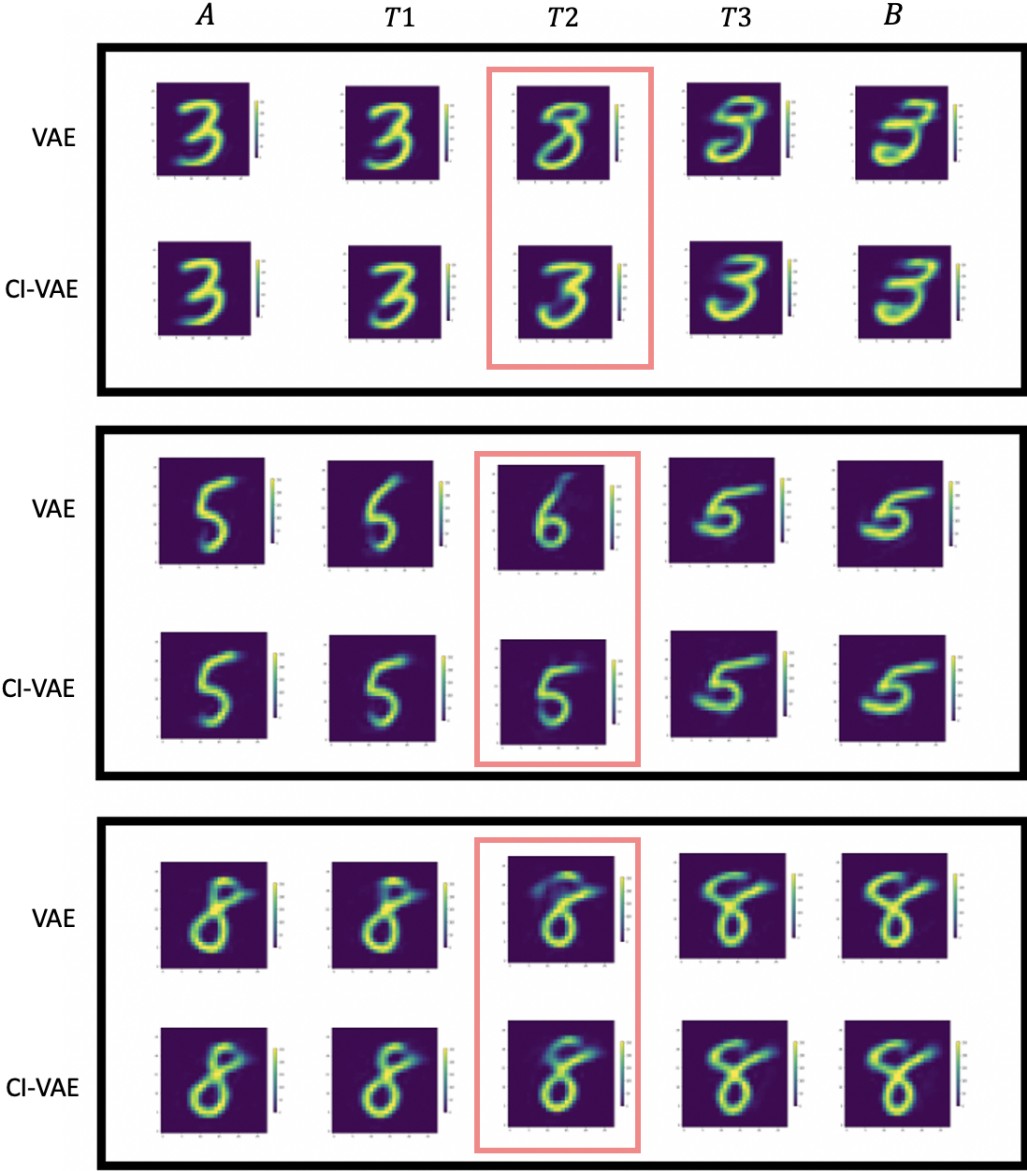

Figure 6: Linear Traversal for three experiments using VAE (top) and CI-VAE (bottom) models. In each experiment, the first and last frames ($A$ and $B$) are real data are also the same across VAE and CI-VAE models. The three middle frames ($T1$, $T2$, and $T3$) are synthetically generated frames during linear traversal for each model. It is shown that for VAE, linear traversal occasionally leads to generating data that may belong to a different class. For example digit 3 is turned into 8 and 9, digit 5 is turned into 6, and digit 8 is turned into 6. However, synthetic data generated by CI-VAE are all robustly within the intended class/digit.

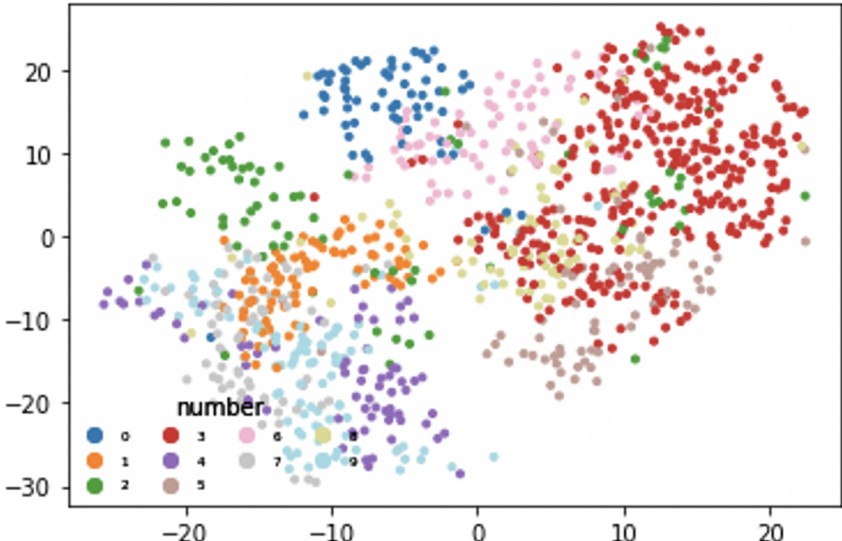

Figure 7: Data augmentation of digit 3 using CI-VAE model shown in 2D TSNE plot on the original dimensions $x$.

is possible that the reconstruction error in VAE could be lower due to the inclusion of the linear discriminator loss in CI-VAE, particularly when the class definitions are not well-aligned with the lower-dimensional structure of the data. By adjusting the value of $\beta$ in equation 3, the contribution of the linear discriminator loss can be controlled to maintain high reconstruction quality in CI-VAE models.

By fitting a logistic regression on the latent space, we have shown in table 1 that the linear separability in CI-VAE has increased by 10%. This is demonstrated in figure 6, where VAE generates images of digits other than the intended one during linear traversal, while CI-VAE generates images consistently within the intended digit class.

The linear separation of classes in the latent space allows for the generation of class-specific data. In figure 7, data augmentation of digit 3 is performed using CI-VAE. In contrast, VAE is unable to generate samples from a specific class because the classes overlap in the latent space. Therefore, in situations where there is a limited amount of labeled data, CI-VAE can be used to generate new labeled samples from any class, potentially improving the performance of the model by increasing the amount of data available for training.

CI-VAE can also be utilized as a semi-supervised learning model Zhu (2005); Chapelle et al. (2009). In such a setting, a portion of the training data with labels is used to train the entire network, while the unlabeled portion of the data is used to train just the encoder and decoder without involving the linear discriminator network.

While CI-VAE effectively addresses the issue of overlapping classes in the latent space and enables class-specific data interpolation, both VAE and CI-VAE may not address the issue of potential discontinuities (holes) in the latent space. There are ongoing research efforts focused on addressing this issue.

## 5 CONCLUSION

In this paper, we propose a new neural network model, Class-Informed Variational Autoencoder (CI-VAE) to enhance within-class data interpolation as well as generate synthetic data from the intended class. CI-VAE is based on the standard architecture of a Variational Autoencoder (VAE), which consists of an encoder and decoder, but it also includes a linear discriminator network. This allows CI-VAE to construct a probabilistic latent space that is linearly separable among different classes, enabling within-class data generation through linear traversal between two arbitrary samples

of the same class. We show that adding a linear discriminator loss to the loss function minimizes the risk of generating samples from a different class during linear traversal. We compare CI-VAE and VAE using the MNIST dataset of handwritten digits and demonstrate that CI-VAE provides more robust within-class data generation and improved class-specific data augmentation compared to VAE. CI-VAE has a wide range of potential applications in supervised or semi-supervised learning problems that require within-class traversal to generate synthetic labeled data.

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
