# A APPENDIX

## A.1 CI-VAE FOR UNDERSTANDING COLON CANCER: INTERPOLATION BETWEEN NORMAL AND CANCER CELLS, USING SINGLE-CELL RNASEQ DATA

Our proposed class-informed variational autoencoder (CI-VAE) can potentially be used to understand the mechanism of cancer development. Here we demonstrate a case study of using CI-VAE for single-cell RNAseq data of Colon cancer Lee et al. (2020). This dataset is composed of RNA expression data of 20K genes for 25K individual cells from colon cancer tissue. 22 cell types are identified in this dataset and for each cell type, I) normal cells, II) borderline tumor cells, and III) core tumor cells are identified. Fig 8 demonstrates a 2D TSNE plot of a portion of these cells.

In this case study, we used CI-VAE to interpolate from normal cells to cancer cells across different cell types to better observe gene expression changes through the development of cancer from normal cells to tumor cells. We selected the top 2K most varying genes (features) across the 25K cells. Cell types assigned to each cell are chosen as the class (22 classes) in the CI-VAE. The CI-VAE model learns the underlying lower dimensional structure of the data while linearly separating these cell types on the lower dimensional space. Within each cell type group separately, we then performed linear interpolation between 100 random normal cells and 100 random tumor cells. The decoder in CI-VAE generates the synthetic RNAseq data for those traversals in the original dimension. Using the trained CI-VAE model, we are able to generate the trajectory of gene expression changes from normal cells to tumor cells, for every cell type.

To validate the predictions made by CI-VAE, we take advantage of cells that are borderline tumor cells. We have shown in many cases, the trajectory generated by CI-VAE results for RNA expression from normal to tumor cells correspond to the trajectory between normal to borderline tumor cells to core tumor cells. Figure 9 demonstrates the gene expression trajectory from normal cells to tumor cells predicted by CI-VAE on the left and the ground truth data on the right for endothelial cell type and for the CD74 gene. Our predictions correctly suggest that the CD74 gene is over-expressed and then drastically under-expressed during the development of cancer. This observation can not be captured if we only observe the state of normal cells and the core tumor cells.

In Figure 10 and figure 11, similar comparisons across different cell types and genes are illustrated. This correspondence between the predictions and the ground truth indicates that the use of CI-VAE on single-cell data may enhance our understanding of the developmental trajectory of cancer cells and thus may enable enhanced drug design, and drug discovery as well as enable detection/treatment on the early stage of cancer.

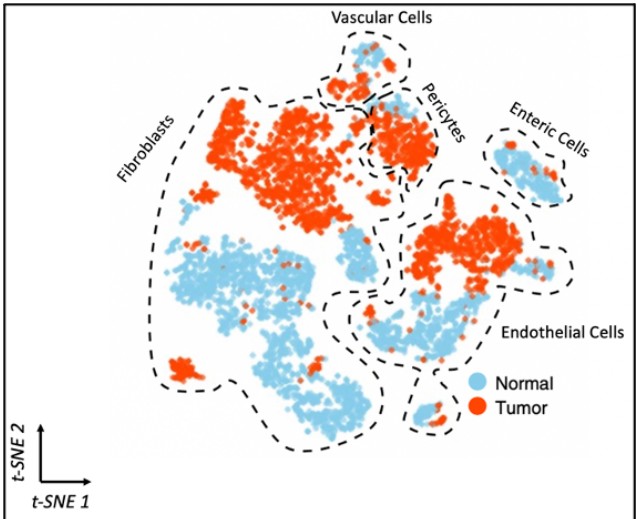

Figure 8: 2D TSNE plot of high dimensional RNAseq data of cells in colon cancer tissue. Each dot represent one cell. Cells are grouped by cell types and blue color represent normal cells and Orange color represent core tumor cells. The figure is extracted from Lee et al. (2020).

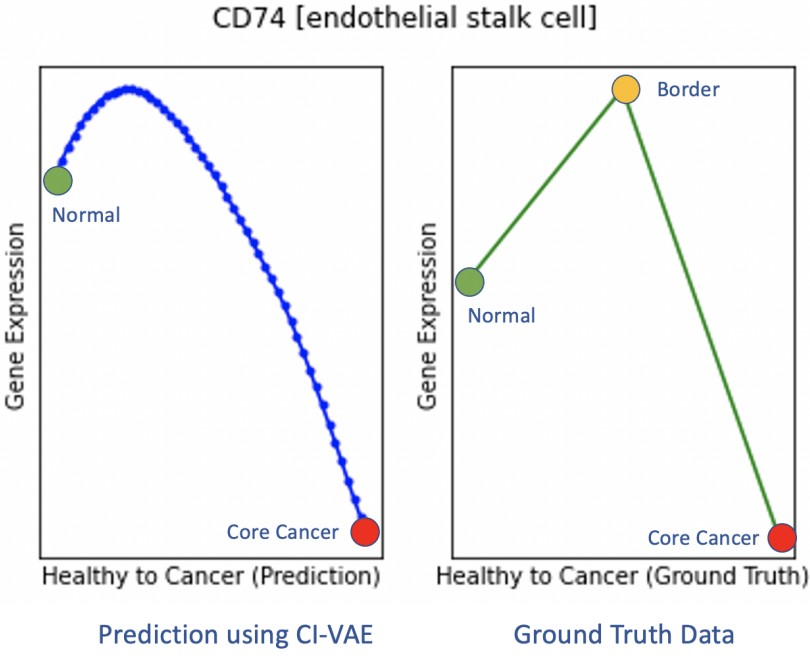

Figure 9: Left, predicted RNA expression from normal cell to core tumor cancer cell using CI-VAE, Right, Ground Truth trajectory with Normal, Borderline tumor and Core Cancer tumor. Results are projected for endothelial cell type and for CD74 Gene. These results are the median of 100 random traversals from normal cells to tumor cells. Our predictions shows that CD74 is over expressed in the onset of cancer development and then under-expressed over the development of core cancer cells.

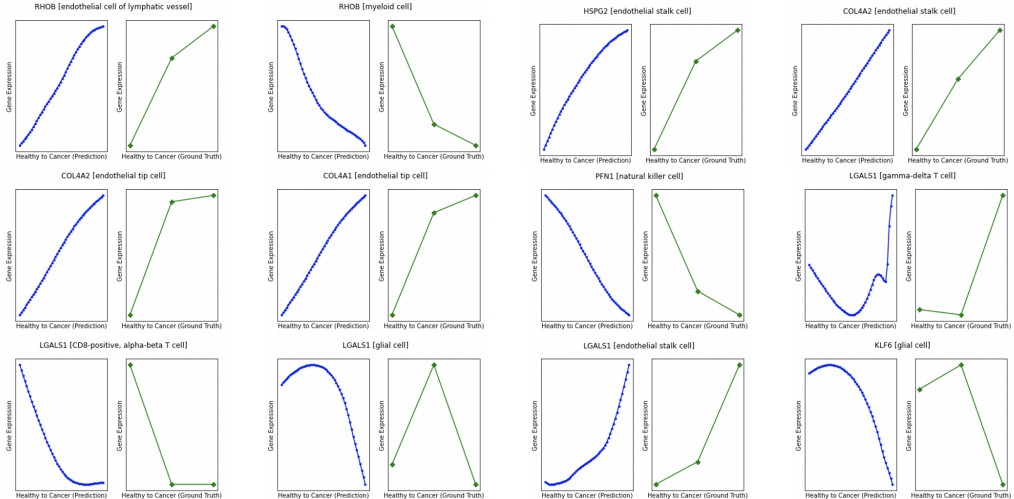

Figure 10: Left, predicted RNA expression from normal cell to tumor cell using CI-VAE, Right, Ground Truth. Each plot pair is for a specific cell type and a specific gene.

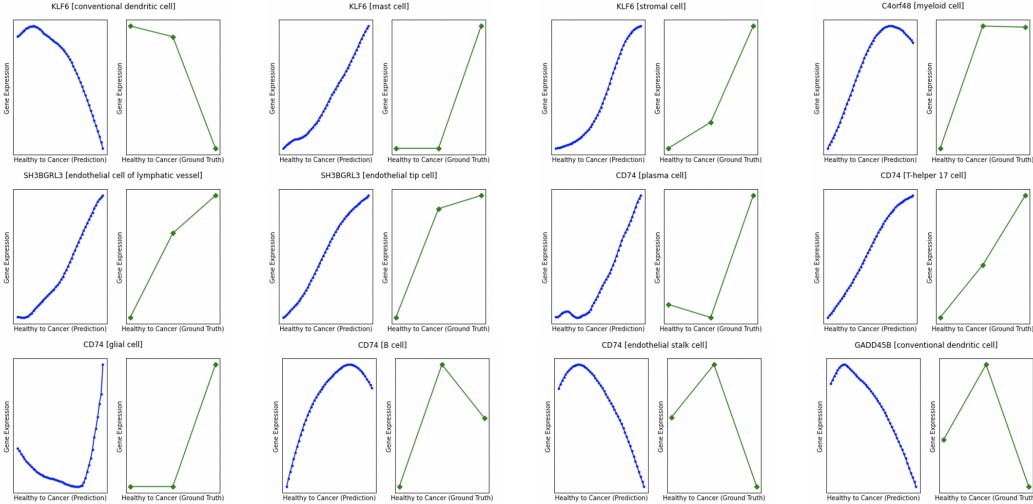

Figure 11: Left, predicted RNA expression from normal cell to tumor cell using CI-VAE, Right, Ground Truth. Each plot pair is for a specific cell type and a specific gene.