# OpenReview forum: "CI-VAE: a Generative Deep Learning Model for Class-Specific Data Interpolation"
_ICLR.cc/2024/Conference — Submitted to ICLR 2024_

### Official Review · Reviewer_Jd7b · 2023-10-31

**Soundness:** 4 excellent
**Presentation:** 3 good
**Contribution:** 1 poor
**Rating:** 3
**Confidence:** 4

**Summary:**

The paper proposes a method to enfore linear separability of classes in the latent space of VAEs. It is investigated on the MNIST dataset and compared to a standard VAE.

**Strengths:**

- The paper is well written and clear and the problem of enabling interpolation and ensuring linear separability of classes in the latent space of VAEs relevant and likely useful.

- The proposed method, at least on the tested dataset, appears to solve the problem of linear separability reasonably well.

**Weaknesses:**

- The method is just investigated on a single and very simple dataset (MNIST) and not compared to other baselines that try to solve the problem at hand.

- While the manuscript references a large number of related works, it does not seem to consider them in much detail. In particular works on the building of generative models where particular considerations are made towards class representations, I would have expected to have been covered in more detail. Such as for instance conditional variational autoencoders or works that focus on learning disentangled representations. There are too many references to cover in detail, but examples include [1, 2].

- "The weights in the model are trained for over 2000 epochs with a batch size of 512 images until convergence is reached (see Figure 3)." - how is this determined?

- While figure 6 is interesting, a quantitative assessment of this problem as well as its solution through the proposed CI-VAE would have been stronger.

- The results on colon cancer mentioned in the abstract and introduction is only included in the Appendix.

- References to Figure 1a-c appear to be wrong?

[1] Kingma, Durk P., et al. "Semi-supervised learning with deep generative models." Advances in neural information processing systems 27 (2014).

[2] Paige, Brooks, et al. "Learning disentangled representations with semi-supervised deep generative models." Advances in neural information processing systems 30 (2017).

**Questions:**

- What is meant by "female hand-written digits"?

---

### Official Review · Reviewer_UN5b · 2023-11-01

**Soundness:** 2 fair
**Presentation:** 1 poor
**Contribution:** 1 poor
**Rating:** 1
**Confidence:** 4

**Summary:**

This work proposes a Class-Informed Variational AutoEncoder (CI-VAE) to interpolate between arbitrary pairs of observations from the same class. The CI-VAE adds a linear discriminator layer to a regular VAE. It can perform class-specific linear traversal and generate synthetic data more robustly compared to a vanilla VAE.

**Strengths:**

1. The manuscript is clearly written.

2. Compared to the vanilla VAE, the CI-VAE outputs latent clusters that are more linearly separable, with a higher linear classification accuracy and a lower reconstruction error. Generated examples also look better qualitatively.

**Weaknesses:**

1. There is no quantitative metric to evaluate between-class interpolation trajectories. How can the linearly interpolated results be validated?

2. The baseline comparison is not sufficient. How does the CI-VAE compare to the other semi-supervised VAE variants?

3. Figures 5 and 6 qualitatively compared 3 examples from VAE and CI-VAE. However, I wonder if quantitative metrics can be applied to compare all samples from VAE and CI-VAE.

4. In Figure 7, it seems that the generated digit 3s are mixed with the digits 5 and 8, so I am not sure if it's practically useful for data generation.

5. I suggest adding key take-aways in the figure captions. It is hard to tell what message that the authors try to convey in some figures such as Figure 5.

6. Minor:

	i. In Section 2, the authors mentioned Figure 1a/1c, but there is no subfigure in Figure 1.

	ii. In Equations (2) and (3), the operation KL should be explained (Kullback–Leibler divergence).

	iii. In Equation (2), putting an arrow above a number is usually used to represent a vector. In multivariate Gaussian, the covariance is a matrix instead of a vector, and thus a bold capital letter $\mathbf{I}$ should be used. The math notations throughout this manuscript don't follow the standard notations, and I suggest the authors to follow the notations in the original VAE paper.

	iv. In Figure 3, the authors should add x axis label (epoch) and y axis label (loss). In text, the authors mentioned that the models were trained for 2000 epochs, but only 700 or 800 epochs are shown in Figure 3. Was there an early stopping criterion?

	v. In Section 3 first paragraph, $28 \times 28 = 784$, instead of $526$.

**Questions:**

1. Code availability: Will the code be made publicly available?

2. Model architecture: Which distance metric was used to measure the reconstruction error? How were hyperparameters ($\alpha$, $\beta$) chosen? How was the number of latent dimensions chosen? Just to make sure, the linear discriminator layer only includes weights, but not biases? What was the dropout rate? These details should be mentioned clearly for reproducibility.

3. Were labels balanced in the training, validation, and test sets?

4. What specific cancer development mechanism can we learn from the interpolation result on colon single-cell genomics data?

---

### Official Review · Reviewer_GLdb · 2023-11-28

**Soundness:** 2 fair
**Presentation:** 1 poor
**Contribution:** 1 poor
**Rating:** 1
**Confidence:** 5

**Summary:**

The paper explores making the latent space of VAEs linearly seperable with respect to class labels. They achieve this by modifying the ELBO by adding a cross entropy loss function to predict the class labels from the latent codes. They show the efficacy of their approach on MNIST an colon cancer genomics data.

**Strengths:**

1. The paper is well written and easy to follow.

2. The idea to learn linearly separable latent space is interesting.

**Weaknesses:**

1. Related work has not been adequately discussed in the paper. I believe learning a structured latent space in VAEs has been a well-studied topic over the past decade but the paper makes no mention of this. This makes it hard for the reader to understand the contribution of this work relative to prior art. For example, a closely related area is learning a disentangled latent space. Some references include Higgins et al., ICLR 2017 and  Mathieu et al., ICML 2019.

2. Building on the first point, adequate comparison to prior work has not been done. For example, I am pretty sure. conditional VAE can generate the same results as figure 6 and will also be linearly separable (just because MNIST is such an easy dataset). It would be good for the author to have some baseline comparisons beyond VAE (which was never designed for this purpose) to convince the readers the benefit of this method.

3. The abstract mentions about application colon cancer genomics data however this is only mentioned in the Appendix. I find those example much more interesting and compelling as to the motivation for learning a linearly separable latent space as opposed to the MNIST examples. I would suggest incorporating this into the main paper, since the authors only use 8 pages, I think there is sufficient space to incorporate this.

4. I find the discussion section too long and winded, in a sense it makes too many unsubstantiated claims of potential benefits of using the proposed method. Here are some examples:

    (i). second paragraph talks about potential benefits of linear classifier vs. non linear classifier, these should be verified to convince the reader;

    (ii).  para 5 talks about tuning the hyper parameters to balance the reconstruction term vs. the cross entropy term, however now quantifiable metric has been provided to understand what are we tuning for? How should we know how to decide what is an acceptable tradeoff between reconstruction quality and classification error (beyond visual inspection of results after training). How does this trade-off translate to linear interpolation along the latent space?;

    (iii). para 8 talks about effective utilization of the proposed method for semi-supervised classification. This is hard to appreciate given the only results in the main paper are for MNIST supervised classification. It would be good to verify this on semi-supervised classification benchmark datasets.

**Questions:**

Please see weaknesses above.

---

### Meta-Review · Area_Chair_biJw · 2023-12-17

**Metareview:**

This paper presents a variational autoencoder architecture that focuses on class-specific information to better learn per-class statistics. The goal is to leverage a discriminator to create linearly separable latent layers, and the method was demonstrated by the authors on both benchmark (MNIST) and real-world (colon cancer single-cell genomics) data.

While the reviewers appreciated the problem being important and the approach being sound and well written, serious concerns were brought as to the novelty of the method, given that many methods currently exist that aim to impose additional structure in the latent space of a VAE. The reviewers pointed out that these other methods were neither sufficiently mentioned nor compared to. Other concerns as to the specifics of the presentation, and other more minor details were mentioned, but this was the main score-driving concern. I therefore do not recommend this paper be accepted.

**Justification For Why Not Higher Score:**

Primarily the concerns on novelty and comparisons to baseline.

**Justification For Why Not Lower Score:**

N/A

---

### Decision · Program_Chairs · 2024-01-16

Reject